# Data Block and Tuple Identification Using Master Index

**DOI:** 10.3390/s20071848

**Published:** 2020-03-26

**Authors:** Michal Kvet, Karol Matiasko

**Affiliations:** Faculty of Management Science and Informatics, University of Zilina, 010 26 Zilina, Slovakia

**Keywords:** data tuple, relational database, index, block, query optimization

## Abstract

Relational databases are still very often used as a data storage, even for the sensor oriented data. Each data tuple is logically stored in the table referenced by relationships between individual tables. From the physical point of view, data are stored in the data files delimited by the tablespaces. Files are block-oriented. When retrieving data, particular blocks must be identified and transferred into the memory for the evaluation and processing. This paper deals with storage principles and proposes own methods for effective data block location and identification if no suitable index for the query is present in the system. Thanks to that, the performance of the whole system is optimized, and the processing time and costs are significantly lowered. The proposed solution is based on the master index, which points just to the blocks with relevant data. Thus, no sequential block scanning is necessary for consuming many system resources. The paper analyzes the impact of block size, which can have a significant impact on sensor oriented data, as well.

## 1. Introduction

Relational databases are still one of the most often used techniques to store data of the information systems. They were firstly defined in the 1960s of the 20th century, but they are still so powerful to cover the current environment and technology demands. They can model any data structure and manage various input data streams. Data values produced by the sensors are often modeled by relational databases, as well. The main advantage in comparison with NoSQL data approach is based on the integrity correlations and transactions [1]. Thanks to that, input data are automatically and effectively evaluated, removing irrelevant, corrupted, or non-reliable data inputs. Moreover, the relational database approach is supervised by the techniques of mathematical apparatus—algebra to reach the performance [2,3].

Performance from the logical point of view is ensured by the data structure and optimization. From the physical point of view, the data block is relevant, whereas, before the processing and evaluation, the particular block must be loaded from the physical database into the memory. Individual blocks, however, do not need to be full, whereas the data are changed, removed, or replaced by new versions. As a consequence, data fragmentation is present, resulting in managing and loading data blocks, which do not hold relevant data for the processing or are even empty.

In the physical database, approved data changes are commonly present. Thus, if you want to get the data from the database, the physical repository is usually contacted to get the relevant data. Ideally, the data image is already present in the memory, but, in principle, it should be identical to the image in the physical database and reconstructable with the support of the log files [4].

As the physical database is separated from user-side access, it is essential to develop sophisticated and, in particular, efficient access to data. The aim of this paper is to propose a complex overview and analysis of the process of data loading during the retrieval process. In our case, data origin from the sensors, providing data to the system. Although nowadays, significant data management stream can be identified, which is based on cloud technology, we still think that there is wide space to optimize existing techniques in existing environmental conditions. One of the reason is based on the current status, there are still many companies, which do not plan to migrate solutions to the cloud, either caused by complex calculations, which would spend too much, another reason is based on the trust and requirements to store sensitive data directly in the company. Companies invest a large amount to the hardware and technology-based, mostly on the Exadata; thus, they operate just local data connections.

Optimization of the query consists of several steps; the most important is just the parsing process and decision-making and how to locate data to ensure performance. This paper deals with own index definition to overcome the necessity to scan all data blocks associated with the data table if no suitable index for the query is present. It highlights the fragmentation and removes its impacts on the performance. It analyses the impact of the data block size definition on the loading process, as well.

Our proposed solution uses secondary indexes as the layer for accessing and locating relevant data blocks. It can be mostly used if data are fragmented on the physical block layer. In that case, free blocks can be skipped without the necessity to load them into the memory buffer cache. Thanks to that, the evaluation is easier and significantly faster. The removal of free blocks is not done automatically in most systems and requires manual operator requests. It reconstructs individual extents, which consist of the defined number of blocks, and interconnection must be ensured. Moreover, such cleaning activity is not executed very often if a significant update stream is characteristic of the system. In that case, free blocks remain in the system to be available for a future update or insert operations [5]. In [6], time-series data are managed as real-time service, where data are located directly in memory. It arises from the problem of relational data evolution over time [7]. Current approaches, many times, model data inside distributed data architecture [8]. A core part of the management is covered by the offline rebalancing structures [9]. Origin of the data can be either relational or object and XML [10]. The impact of storage space configuration is discussed in [11]. Fragmentation problem and consecutive solution methods on the logical scheme are proposed in [12]. It uses cloud architecture, but the principles can be applied in stand-alone systems, as well. Fragmentation on the index layer is defined in [13]. Namely, if the data are changed, appropriate indexes must reflect these changes, mostly by adding new nodes to the index tree. It means that the size and depth of the index are still rising, forcing the system mechanisms to ensure index balancing. To get rid of free nodes, which do not hold relevant data and can be grouped together with another one, rebuild techniques of the index are available. Their principles are described in [14] based on synchronization, replication, and allocation in [15]. Defragmentation can be done either offline or online. In the case of using an offline rebuilding process, a particular index is not available at all, whereas the process is physically implemented as a drop followed by new index creation [13]. Online transformation is more often preferred, in current research, as well as the commercial environment, whereas the original index is still available until the new, more effective index is created. Afterward, the original index is dropped and replaced by a newer version.

Most of the proposed solutions are, however, mostly based on the upper level, reflecting data distribution and fragmentation. In our solution, we manage data access on the lowest level of the processing inside the physical blocks and access techniques. We do not deal with data defragmentation and block merge, whereas it would require too many resources, and often, a very short time later, new blocks must be allocated. Therefore, we just do not use them for the query evaluation, but they are still present and associated with the table segment.

## 2. Client-Server Architecture

The connection between user and server is created via the listener, which contacts the particular process monitor background process. It creates the server process on the server-side and interconnects it directly with the user process. Afterward, the communication is done in a straight manner, without the necessity of any other system cooperation. Figure 1 shows the architecture of the server and client connection. Naturally, data cannot be operated from the client directly due to many reasons. The most significant aspect is related to the security, reliability, and integrity of the whole system. Data management, individual access methods, and transfer are protected by the background processes located in the instance of the database server. The instance itself consists of two main structures—memory structures and background processes. Our proposed solution is placed between the instance and database itself to monitor and manage the process of data loading into the instance memory, where individual blocks are evaluated and sent to the client in the form of the result set. Thus, if the client requests some data to be produced, in the first phase, particular data must be identified on the server followed by the loading process (data blocks are transferred from the physical database storage into the memory, where the blocks are parsed and data tuples extracted). In the last phase, the result set is composed and proposed to the client.

In the ideal conditions, data can be located using the index. If no suitable index is present, currently, the system is forced to scan block by block up to high water mark symbol, characterizing the pointer to the last block associated with the table. As a consequence, the whole process is time and resource-demanding.

## 3. Index Structures and Access Methods

One of the main property of the query processing optimization is just an index structure reflection. The index itself is used for direct access to the row inside the database by using ROWID on the bottom leaf nodes. ROWID is the locator for the data and consists of these layers: identification of the data file, in which the row resists, the pointer to the block, and position inside it. Moreover, it uses the specific object identifier, as well. Thus, based on the definition, the ROWID value is unique for the standalone database [16].

An index is an object of the database with the associated tablespace, whose aim is to monitor data and reflect changes inside to be prepared for the query to access the row using ROWID directly. In the database systems, various index structures and approaches can be used. The most often used is just the B-tree and B+tree [16], whereas it maintains the complex efficiency despite frequent changes of records. It does not degrade over time and remains balanced. The structure consists of the tree in which each path from the root to the leaf has the same length [17]. Three node types are present: root, internal node, and leaf node. Root and internal node contains pointers *S_i_* and values *K_i_*, the pointer *S_i_* refers to nodes with lower values than the corresponding value (*K_i_*), pointer *S_i+1_* references higher (or equal) values. Leaf nodes are directly connected to the file data (using pointers) [18].

The model of the B-tree index structure is in Figure 2. The leaf layer contains locators of the rows in the physical database—ROWID values.

B+tree index approach is an extension on the leaf layer, where individual nodes are chained together, forming a linked list. Thus, such a layer holds sorted data based on the attributes, which are there indexed [16,17].

Other index techniques mostly arise from this category and improve either modeled data group or data shapes to be processed, like inverted B+tree key, its unique version, table, or cluster index. Different structural approaches are based on a bitmap or hash indexes, which cannot be, however, universally used due to specific requirements and limitations [16,17,18].

In the previous part, we supposed that data are heap organized with the index enhancements to access data. Index organized tables (IOT) fuse data and index layer into one compact module. In that case, on the leaf layer, no pointer to the data is present; however, direct data blocks are there. The structure of the IOT is delimited by the primary data and non-key column data stored in the same B+tree structure. Thus, data are stored within the primary key index. The main reason for developing such a structure is interconnected to efficiency. Accessing data via primary key is quicker as the key, and the data reside in the same structure. There is no need to read an index and then read the table data in a separate structure. Moreover, whereas the primary key cannot hold duplication, total storage requirements are reduced. No ROWID value is stored, as well [19].

In our solution, however, we do not use IOT structure principles but prefer data and index separation for many reasons. In the past, many times, virtual values characterizing primary keys were used, mostly provided by the sequence and triggers to limit the usage of the composite primary keys, whereas many attribute connections had to be done. Practically, table joining uses just one attribute. Such definition has, naturally, many negatives, and to obtain original primary key values, tables have to be joined together. Moreover, such values have to be unique, thus physically implemented constraint by adding a new index is used. Many other checks have to be done, as well, to ensure data reliability and consistency. Secondary indexes in IOT are strictly dependent on the main (primary) structure, whereas they reference individual primary key leaf nodes. As a consequence, the effectivity of the structure is enhanced by the primary key index size and its reconstruction over time to limit non-appropriate index nodes.

Therefore, we have decided to use the original principle separating index access techniques and data. It offers a universal solution, whereas, in principle, the primary key can be absent. Our proposed solution is not strictly directed to use the primary key, but any suitable index can be used. Thanks to that, a more generic solution is present, whereas the system can dynamically select the best appropriate index to access data based on the current environment and index loading. In comparison with IOT, a wide range of indexes can be defined as the master to locate the data.

Query processing consists of several steps, which are consecutively executed. The output of the individual step is transferred as the input of the subsequent one. Figure 3 shows the query processing steps. Parser performs syntactic (command grammar) and semantic (object existence and access rights) analyzing of the query and rewriting the original plan to a set of relational algebra operations. Optimizer suggests the most effective way to get query results based on the optimization methods, developed indexes, and collected statistics. Thus, it selects the best (suboptimal) query execution plan, which is used in the next row source generator step. It creates the execution plan for the given SQL query in the form of a tree, whose nodes are made up of individual row sources. Afterward, the SQL query is executed with emphasis on the provided execution plan. The result set is constructed and sent to the client [1,2].

The most important step in terms of processing efficiency, optimization, and access rules is just the execution plan, which determines the usage of indexes. Access methods can be, in principle, divided into two categories—direct access to the data files (table access full (TAF) method) or access by using an index (index scan) or just by their combination. TAF traverses the entire table, all data blocks associated with the table, which can be physically widespread into multiple data files. The word entire table is significant. For each data segment, a high water mark (HWM) symbol is defined as determining the last block for the particular structure, which can be generally empty, whereas new blocks are not associated separately, but as the group forming extents. That means that also blocks with no valid data must be moved into the memory for the evaluation. Even more significant limitation is formed by the fragmentation properties. Block does not need to be completely full; in the real environment, there is significant fragmentation on the block granularity caused by the variability of individual row size, as well as processes of data changes, where the updated row does not fit the originally allocated space [20,21,22]. To ensure efficiency and robust performance, it is necessary to limit the usage of TAF methods.

The aim of the index definition is to remove such an impact. Index for the primary key and unique constraints are defined automatically, while others are user-provided. The index can significantly improve the performance of the query, but there is slowdown during data modification operations, whereas the change must be applied inside the index, as well. Therefore, it is not effective and even possible to define all suitable indexes [23]. As a consequence, whereas no suitable index is proposed, TAF methods are repeatedly used, resulting in poor performance and user complaints.

The index scan method category searches the data based on the index. The output of this method can be either whole data set if all of the required attributes are present in the index or set of ROWIDs, which are consequently processed, and particular blocks are loaded into the memory by using the ROWID scan method. Following index scan, types can be distinguished:Index unique scan—based on the condition, which always produces unique data, thus either one or no rows are selected.Index range scan—standard method, in which the index columns are in the appropriate order, but there is no guarantee that the result will be no more than one record.Full index scan—the whole index is searched in a sorted manner, and particular ROWIDs on the leaf layer are selected. The condition on the leaf layer can be directly evaluated.Index skip scan—method, in which the leading index attribute is not suitable, but the rest ones are appropriate. In that case, it works like the index in the index; thus, the first index attribute is skipped.

The point of the processing is, therefore, the suitability of the index. If the order of attributes is not suitable, the index is not used. Let’s have a table T consisting of four attributes: A, B, C, D. Let’s have an index I formed by the pair of the attributes A, B. If the query requests values of the attribute C based on the attribute D, it is clear that the particular index cannot be used. Thus, the TAF method is used. The only solution to cover the problem is to create a new index, which has, however, negative aspects in the term of the change management performance. If the system is dynamic with various query construction types over time, the problem is much deeper. The whole table must be scanned sequentially with an emphasis on the data fragmentation. Therefore, performance is getting worse and worse. Deleting old records does not solve the problem, too, whereas the number of blocks allocated for the object is never decremented (HWM cannot be shifted to the left part of the linked list). The point is, therefore, clear—propose the solution to cover the problem by removing the impact of full table scan necessity. Next section deals with our proposed solution.

## 4. Own Proposed Solution

A characteristic feature associated with the usage of the full index scan method category is the direct evaluation possibility. Although the order of the attributes inside the index does not fit the query, the relevant attributes are present, however, in non-suitable order. As a consequence, if the ROWID on the leaf layer is selected by passing the Where clause condition of the query, it is certain, that the record will contain the data needed to create the result set. Thus, no irrelevant data block is loaded into the memory buffer cache, except for the migrated row problem. Therefore, if the evaluation cannot be done on the index layer itself, the system transfers the processing by using the TAF method instead.

Our proposed solution uses a different principle. If enabled, the index will be used in any case. Thus, if the order of the attributes is good for the index access, original methods are used. Vice versa, if no suitable index is present in the system, instead of scanning all blocks sequentially, the proposed solution uses the index on the leaf layer. It is used just for the location of the blocks with relevant data, whereas after many data operations (insert, update, and delete), fragmentation is present. The importance of our solution definition is described in the following example. Let’s have four data rows for the table. For simplicity, let’s assume that each data row is located in the separate data block at the beginning. Then, insert two new rows, which will be located in the same data block. Notice that the blocks are associated with the object in the form of individual extents and not the blocks directly. So, let’s assume that the extent contains two blocks. Thus, after the execution, six blocks will be used, and the last one will be empty. Now, in the third step, remove the data of the third tuple. What about the results? The third block will be associated with the table but will be totally empty. It is clear that only four data blocks are relevant for the evaluation, and they only contain the same data portions. By index, they are accessible via ROWID values of the index. However, in this case, if the index does not contain the attributes characterizing the query condition, the TAF approach method is used. Unfortunately, the TAF method does not have any information about the empty blocks associated with the table, and no data defragmentation or migration is done due to performance impacts—such a table would be inaccessible during such a process, which is not acceptable. Moreover, nowadays, the number of update statements is high and is still rising; thus, data consolidation would require to be executed too often to ensure the benefit, but it is too resource demanding. Overall, the improvement would be minimal, even if any. Thus, by using TAF in the described situation, six blocks would be loaded into the memory, but two of them do not provide any data. The global efficiency would be 4/6—just a bit higher than 66%. Sure, it is just a simple demonstration of the problem; in the real environment, performance significantly below 50% would be reached, so more than half of the system work would be unnecessary for the evaluation and the processing. This is, of course, a huge problem in terms of the performance and growth of the data requirements and complexity. Individual processing steps are shown in Figure 4. The black color of the block represents its occupation, and white blocks are empty.

In this paper, we propose own solution based on the master index. It uses the fact that each data row in the relational database can be uniquely identified using the primary key; thus, each table usually contains a primary key definition, which has the property of the uniqueness and the minimalism. Whereas the value of the primary key must be present from the definition (cannot hold undefined NULL value), and the primary key automatically creates the index; in the system, at least one index exists with ROWID pointers to each data present inside. From this point of view, if the index would be used, just the relevant blocks would be selected. The solution is shown in the data flow diagram in Figure 5. When the query is obtained to be evaluated, first of all, existing index suitability is evaluated. If there is a suitable index, naturally, it is used. In this case, therefore, there is no change compared to existing approaches. One of the index scan methods is used, either index unique scan, index range scan, or index full scan with its variants. If there is, however, no suitable index based on the data characteristics and conditions, own proposed solution as the data optimizer extension is used. The system evaluates whether there is the master index definition for the particular table. If not, the TAF method must be used with all its limitations. However, if one of the indexes is so marked, it will be used, not for the data evaluation, but just as data access.

### 4.1. Master Index

There is only one strict requirement for the master index definition (MID)—all data rows must be accessed via it. Thus, it must cover all the data. As mentioned, the most often relational database index structure is B-tree and B+tree. It has one limitation—undefined (NULL) values are not indexed. So, if at least one of the indexed column has the property of potentially holding NULL value, there is no certitude that all the data are present by using the index. Whereas, in principle, each table is delimited by the primary key definition, and such a suitable index should always be present. The point is just whether it is the best suitable or not.

The master index definition is defined for each table and can be selected either automatically or manually based on the user’s decision. The decision for the table can be selected this way [24]:Alter table <*table_name*> set MID = <*index_name*>;

In the previous case, the user defines the master index manually. Thus, if the index is dropped or denoted as corrupted (needs to be rebuilt), the MID parameter is automatically set to NULL, and the proposed technology will not be used later. Thus, if the setting for the table would be NULL, the proposed extension would not be applicable for the table, resulting in using the original TAF method.
Alter table <*table_name*> set MID = NULL;

The selection of the master index can be done automatically by the system, as well. The decision is done by the optimizer based on the current statistics of the index and the whole system. Suitability of the index is to be declared by its size on the leaf layer. Generally, the fewer amount of nodes indicates better performance. Another aspect of the selection is just the availability of the index in the buffer cache memory structure. Similar to the table, the index must be loaded into the memory to be processed, as well. If some index is already available there, either partially, the process of the loading using I/O operations is removed and shortened. Therefore, it is gainful to use automatic system management and decision-making. The option is done on the table granularity by using the following command:Alter table <*table_name*> set MID = AUTO;

The advantage of this approach is reflected by efficiency. If some index is dropped, the system automatically evaluates whether it is marked as master or not. If so, a new suitable index for the processing is selected, if possible.

### 4.2. Index Master Method

In the previous paragraphs, the principle of master index definition selection is described. For now, it is necessary to explain the principle of data access. The TAF method principle is characterized by the sequential scanning of all data blocks associated with the table. It uses the fact that the individual blocks are formed in the extent shape, which is linked together [24,25,26]. As described, for the processing and evaluation, the block is always loaded into the memory, even if it does not include relevant data for the query and if it is empty. Thus, in the first phase of the development of own approach, the aim is to remove such blocks from the evaluation. Our firstly defined solution is based on two sides linked list. Each block then consists of the information about the fullness of the direct following block (model 1). Thanks to that, the empty block is skipped from the evaluation. The disadvantage is the necessity to store the two-way linked list and modification of the whole path after the change on the block level, as well. Our second proposed solution (model 2) has improved the original approach by storing the pointer to the next used data block. In this case, the two sides linked list is used. The principle is shown in Figure 6. Black block is occupied, while white is free. Let’s assume that the third and last associated blocks are empty. Bold arrows indicate added pointers to the other blocks. While the last block is free, the fifth can point either to the same (fifth block) or NULL pointer that can be used regarding the consecutive way of evaluation. NULL is better from the size point of view but worse for the management.

Another model to be evaluated is based on the fullness of the block located in the data dictionary (model 3). Current database systems store information about the fullness of individual blocks associated with the segment in the data dictionary. There are, however, just four categories reflecting the free space. The first type covers the range from 0 to 25% of free space, the second is from 25% to 50%, the third reflects the range from 50% to 75%, and the last from 75% to 100%. From the defined category types, it is, however, impossible to determine blocks, which are totally empty. Extending the solution by explicit identification of free blocks by their positional location (pointers) solves the problem only partially, whereas it creates the bottleneck of the system. Data dictionary would be necessary to locate block, not just in case of identification of the free space available for the new row, but it would be an inevitable part for the queries, as well. As evident from the results, it brings only a slight performance benefit because of the locks operated on the data dictionary to ensure holding correct data. In the Experiment section, it is expressed as model 3.

After the complex experiments, we come to the conclusion that the proposed solution is robust but not optimal. Inside each block, specific space has to be allocated for the pointers and fullness management. As a consequence, each block itself has to be shortened based on the size. Therefore, we define also another solution (model 4), which is just based on the master index. It is used as the source of pointers to the data blocks. Is it possible to determine existing and at least partially occupied blocks? Sure, it is, by using ROWID pointers at the leaf layer. To ensure efficiency, we add a new parameter associated with the MID. It holds the pointer to the first node at the leaf layer of the index. Thanks to that, it is not necessary to traverse the whole index from the root. The name of the parameter is MID_pointer_locator and is maintained automatically; thus, if the structure of the index is changed as the result of rebalancing, such a parameter is automatically notified to ensure correctness.

MID_pointer_locator gets the first index node for the processing and the first ROWID pointing the block inside the database. B+tree has a linked list on the leaf layer; therefore, individual data segments can be directly located from that level. Logically, there is a list of ROWIDs, which is used to access the physical data. Thus, non-relevant data blocks are not processed at all, whereas no ROWID points to them.

Our proposed approach uses the private global area (PGA) of the server associated with each session separately. In this structure, local variables are stored. In our case, we use it for the list of individual blocks. Multiple rows can be located in the same data block. Therefore, before the evaluation, the address of the block is extracted from the ROWID value, which is consecutively checked, whether such block has already been processed or not. Notice that the whole block is evaluated, not only the row itself. The reason is based on the efficiency of I/O operations. It could happen that block with multiple records is read into the memory. If only one record is evaluated, such a block would have to be processed later for further records. In the meantime, however, the block could be removed from the buffer cache as it is a clean block type, and no changes are made to it. Thus, the number of I/O operations would increase beyond the number of blocks actually used. The diagram expressing the processing steps is shown in Figure 7.

### 4.3. ROWID vs. BLOCKID

In the previous definition, the principle of using ROWID values to identify blocks is used. As described, management is extended to check whether such a block has already been processed or not. The aim of the solution is clear, to minimize the amount of I/O operations, which is part of the most expensive operations of the systems themselves. As a result, it would be grateful if the solution can use identifiers of the block on the leaf layer instead of the ROWIDs. It is, however, not possible directly, whereas there is no possibility to modify existing index approaches in the core of the database system. The solution is, therefore, based on two interconnected index structures. One of them resides the original and consists of the ROWID values in the leaf layer. The difference is that they do not point to the data blocks in the physical database but are routed to the second index. Pointers are always paired, from the index to the block module and vice versa as well. Thanks to that, any change in the block management can be easily identified, and the whole supervising layer can be notified. Block module form is similar to the index; it uses the B+tree structure too. On the leaf layer, pointers to the physical database are on the block granularity. If any block is freed and is associated without particular data, such blocks are not part of the block module and automatically skipped. The master index method uses only the block module and scans the blocks in a parallel manner. If there is any change in the data, the original index is used, which, however, automatically reflects the change in the block module, if any change in the segment or extend block positions are done. Select statements use direct access to the block module. The architecture of the solution (model 5) is in Figure 8 [24].

### 4.4. Architecture Performance Evaluation

Performance characteristics have been obtained by using the Oracle 19c database system based on the relational platform. For the evaluation, a table containing 10 attributes originated from the sensors is used, delimited by the composite primary key consisting of two attributes. The table contains 1 million rows. No specific indexes are developed; therefore, the primary key is denoted as the master index.

Experiment results are provided using Oracle Database 19c Enterprise Edition Release 19.3.0.0.0—64 bit Production. Parameters of the used computer are:Processor: Intel Xeon E5620; 2.4 GHz (8 cores),Operation memory: 48 GB DDR 1333 MHzDisk storage capacity: 1000 GB (SSD).

The size of the row is dynamic, based on the data flow and types. The range is from 98 bytes up to 145 bytes, and the average size is 129 bytes. When using mathematical calculations, block data requirement should be 16,250 blocks, if using 8 KB size of the blocks, which reflects approximately 2030 extents, if each extent is defined by eight blocks. Environment and properties are not, however, ideal and fragmentation if present in a range of 50%. Thus, the total number of blocks used in the system is 32,547, created as a dynamic result of data updates over time. The 20% of the blocks are totally free but are not disposed of due to dynamic update operations and assumptions for the future usage of them. The 25% of the blocks are not loaded more than one half. Only 7% of the blocks are almost full (filled more than 90%).

As described, the referenced table consists of 10 attributes, and five of them are selected by the query. The selection of individual attributes is dynamic and depends on the experiment properties. In principle, two attributes are part of the composite primary key as a unique identifier, and those are not selected because of the evaluation of usage non-key index. The attribute selection is done once for the whole round of the experiments. Each experiment is evaluated ten times, and the results express the average values. The reason for random attribute selection is based on the difference in data distribution, characteristics, data types, and operations. Principles are discussed in [23,24].

Results are provided for the five models, and a full table scan (TAF method) is used as well as the reference. Model 1 stores in each data block information about the fullness of the direct following block. Thus, the free block can be relatively easily identified. On the other hand, it is still necessary to load each block to obtain information about the next block. Thus, the limitation of the solution is based on the group of free blocks, in which each of them must be transferred to the memory for processing. However, it is not necessary to evaluate the block data themselves if the block is free, which simplifies the process. Model 2 removes the identified limitation by storing pointers to the next block containing relevant data. Thanks to that, free blocks are skipped from the evaluation process. As shown from the results, it has brought only small performance benefit. Model 3 stores information about the fullness of the individual blocks in the data dictionary. In this case, the management limitation can be identified caused by the necessity to move blocks between categories, when changing. Moreover, it must be always evaluated whether the block is entirely empty or not, which can be complicated in case of dynamic sensor data flow.

Model 4 is based on our proposed master index approach. The management does not require any additional structure, and data are always consistent with no necessity to extend background instance processes. As described, it is based on the index, which is always actual, reflecting the changes performed on the data set. Model 5 adds a new layer by modeling data access on the block granularity. The principle is described in Section 4.3. The index is extended by the block identification rather than the data tuple itself.

The results are shown in Figure 9, expressed by the processing time in the second precision. Values in the graph express the improvement or slowdowns of the processing time in percentage. The referential model is based on the original method for the data access, if no suitable index is present in the system—TAF method. Based on the results, it is clear that all solutions provide significant improvement. Ten percent of the data is part of the result set, and there is a fragmentation level of 50%. In model 1, the performance requires just 23.5% of the processing in comparison to the reference model TAF. The model 2 performance is even a bit better and requires only 21.3%, whereas irrelevant blocks can be skipped with no transfer and loading process into the memory. Model 3 is based on the data dictionary and provides similar performance results (22.4%). Our proposed solutions expressed in models 4 and 5 are the best. Model 4 is based on the original master index method. In this case, the processing time requires only 17.3%, assuming that the selected index is not already loaded into the memory. If so, the processing time can be reduced by 5.1 percent. The last model (Model 5) provides the best solution in comparison with other models. It adds a new layer for dealing with the data blocks themselves. In this case, the processing time percentage is 12.1%. Notice that in the optimal environment, no fragmentation is present. In that case, individual solutions would reach almost the same results. However, in the real environment, data evolve, are changed, and historical data are removed from the system, whereas they are volatile and do not provide additional information value [27,28]. As a consequence, data blocks are fragmented and can be even free. Based on the results, if the fragmentation is present, the location of the relevant blocks by using the master index brings significant benefit. If the fragmentation is 50%, performance expressed by the processing time can be lowered up to 12.1% in comparison with the TAF method. It is caused by the reduction of the I/O transfer, which is the most demanding operation of the query evaluation.

The performance of the system is, however, not just the processing time itself. It is clear that there is a significant benefit to the proposed solutions. On the other hand, it is useful to monitor the total costs in a complex manner, as well. For these purposes, the second part of the evaluation is based on the size of the structure necessary to reach the results. For illustrative purposes, values are expressed in percentage. Model 1 reduces the available size of the block holding the data tuples, whereas it references the next block by storing its fullness. Additional demands are 5%. Model 2 uses only pointers; if the block is totally empty, it is skipped from the evaluation and consecutive processing. It is significantly beneficial if the free blocks are grouped together. Thus, the additional demands are 2%. Data dictionary resources only extend the existing categorization by the definition of totally free blocks (Model 3). It requires less than 1% of the additional demands. Model 4 does not add any new structure; therefore, there are no additional size structure requirements. Model 5 is different. It provides a significant benefit in the processing time aspect; however, a new layer dealing with the block pointers must be added. It consumes an additional 13%. Results in a size manner are shown in Figure 10. The reference model uses the TAF method. Notice that from the relational paradigm itself, each data table should have the primary key definition; thus, one index, which can be denoted as master, should be always present. If several indexes are present, master index technology can provide even better results.

Experiments evaluating fragmentation and rating of occupation can be found in [25]. Based on the reached results, proposed solutions in models 4 and 5 work effectively, only if the fragmentation is at least 15%. If not, then the original TAF method would provide almost the same results.

## 5. Database Instance and Processes

Database instance consists of several memory structures, which store and manage data in a temporary manner, both before saving changes directly to the database, but also in the process of retrieving data from the database. It has several structures for dealing with metadata, as well. In the following section, the most important parts of the data retrieval process are mentioned and described.

### 5.1. Buffer Cache

The buffer cache is the most important memory structure when dealing with data management and retrieval. During the data values changes (update process), it is necessary to load data from the physical database storage into the memory to become data available. Thus, the data of interest must be copied into the memory buffer cache before the processing itself. Changes (such as inserting, removing, or updating) are always applied to the blocks located in the memory and consecutively reflected in the physical data blocks in the database. Security of the transactions and durability option is ensured by the transaction logs, which are enclosed by the UNDO tablespace in the storage system, not in the memory [1].

Data of any query are cached before the composition of the result set in this structure, as well. From the physical point of view, it is implemented as the matrix of the blocks containing rows of interest. The block is by default defined by the size of 8 KB, and it is fixed-size. Thus, the transfer from the database into memory is defined by the block granularity, and no smaller data portion can be loaded. In principle, the block size can be either default (8 KB) or as the power of the 2 KB size.

The size of the buffer cache structure is, therefore, crucial. From the performance perspective, there must be enough space to have frequently accessed data blocks directly loaded. Nowadays, fortunately, most database systems can automatically manage and distribute available memory into the individual structures and rebalance them on demand.

### 5.2. Log Buffer

The log buffer is a small memory storage place, where transaction information about the performed changes are temporarily stored before they are written to the redo log located physically on the disk. Data about data modification are called change vectors. Thanks to that, no data can be lost, if any data portion is to be updated or modified, and change vectors are created and stored sooner. Database system instance processes ensure that before committing transactions, all particular logs are physically stored. The reason for creating a log buffer memory structure is based on the performance. Writing and accessing memory is far faster in comparison with I/O operations on the disk storage. On the other hand, the system tries to transfer data from memory into redo log storage almost immediately. It is, however, performance benefit, whereas, in the system, many transactions are present in parallel. Particular change vectors are composed of the batches and loaded in that manner. From the physical perspective, it is defined as the circular buffer.

Significant limitation and performance disadvantage of the changes logging is reflected by the select statement. Each data to be updated must be loaded into the memory buffer cache. When executing, changes are reflected in the data block in the memory. Thus, in the physical database storage, the original data block is located, and change is performed on the memory block, regardless of the transaction activity and status. It is clear if the transaction is approved, a particular block contains current valid data. Unfortunately, if the transaction is refused, particular data changes must be rollbacked, as well. Similarly, when querying data, a consistent image must be built [1,2]. For these purposes, log data, either from memory, but from the physical storage, are used. Active transaction log data can never be lost and rewritten, mostly due to transaction rules, consistency, and durability. The process of the block reconstruction can be, however, significantly resource consuming. Figure 11 shows the whole process. In the first phase, “block info” is obtained, followed by access to the block itself. If it is current, it can be directly processed, forming a result set. The more complicated situation occurs if some transactions are applied to it. In that case, log data must be used, aiming at the composition of the current image. Unfortunately, data about the approved transactions can be overwritten; thus, if the active transaction lasted too long, it can happen that the original image cannot be created. As a result, “snapshot too old” exception is raised and produced to the user interface.

Such a situation is very demanding in the complex query processes, like analytical queries, prognoses making, etc. The problem can be partially solved by using archive log repositories. As shown in Figure 12, log data from the physical storage can be copied to archive repositories before replacing them with new data change vectors. Thanks to that, it is always possible to reconstruct an image, although it can last too much and consume many resources. Figure 12 shows the architecture of the archiving and the process of obtaining the image from the archive log mode. In comparison with Figure 11, it accesses the archive module using the archiver background process, if the appropriate image cannot be constructed from the live active (online) data logs. The negative of the mentioned technique is associated with disk storage demands. The size of the archive log mode repository is still rising. Therefore, it is necessary to cover the physical transactional data management, as well, to ensure the robustness and efficiency of the whole process.

As evident, our proposed solution is based on the data block architecture. Its aim is to access the block with relevant data as soon as possible by skipping inappropriate or free data blocks. It is, however, inefficient just to cover the process of the data location without the reference on the direct data stored in the table with an emphasis on the running transactions. Therefore, we propose our own solution for the transaction layer, as well. Three models are proposed, categorizing the efficiency, performance, and usability of the whole process. The first model (M6) does not change the architecture of the transactional log management in the archive data layer but takes emphasis on the relevance of the archive data. We call it the “suitability identification model”, and its aim is to remove too historical archive data, which are not relevant for the image composition anymore. Any data change vectors applied to the data later than the beginning point of the currently active transaction cannot be removed. Data about the transaction that started before the beginning point but ended after such point cannot be removed, as well. Data characterizing current active transactions cannot be removed from the definition. They are either relevant for obtaining data snapshots, as they were in the past, but are the crucial substance of the transaction definition, too. Figure 13 shows our proposed solution model. Log data are not categorized based on the storage location (memory, redo log in the database, archive repository), but with reference to the time and transaction definition.

Figure 13 shows the principles of expired archive data identification. It is based on the minimal time of the beginning point from all active transactions. All data logs, which are older, can be removed from the system, whereas it would never be necessary for original image construction.

The proposed suitability model does not solve the performance aspect of the whole solution; its aim is to secure the disk storage capacity just with relevant data.

The second model (M7) is based on the concept defined as M6 but adds a new layer with the temporal pointers. Each data block on the physical storage is defined by the BLOCKID values (defined in Section 4.3). The principle is similar to ROWID, but it references only block granularity. In the extended transaction layer of the data dictionary, pointers to the changes applied to the particular block identified by BLOCKID are stored. Notice that the individual changes are defined in the linear linked list shape, and sorting is done by using the identifier of the transaction and time denotation.

As mentioned, the transaction pointer layer is modeled by using a linear linked list. From the physical implementation, it can be defined either as a varray, nested table as the table attribute or by the table itself. Based on the performance evaluation, the most suitable solution provides the direct table [17,18]. The reason is mostly based on indexing. Sub-tables covered by the attributes cannot be a part of the index structure; thus, the searching strategies are strictly limited to sequential scanning of the linked list. Although data stored inside are sorted, methods of bisections cannot be used, resulting in the necessity of using a full table scan on the nested table granularity. Model M7a is characterized by the varray, model M7b uses the nested table, and model M7c uses the real table for modeling transactional references to the logs.

The last model (M8) used in the performance evaluation strategy is proposed by us, as well. It uses temporal versioning, characterizing individual changes on the transaction management layer. It shifts the problem into the physical data storage layer. Thanks to that, data access can significantly benefit, resulting in offering a robust, reliable, and fast solution. The principle is based on the composition of the temporal groups. Each temporal group is associated with one transaction and consists of the data, which are influenced by it. Thanks to that, relevant changes are located in the same group, consisting of one or several data blocks in the physical database storage. Thus, data about one transaction are physically located nearby. It reduces reading requirements and minimizes I/O operations. Naturally, individual change vectors can be either embedded in the system (model M8a) or references using pointers (model M8b). Referencing is done by using the same technology as the index ROWID principle uses—direct pointer to the relevant data located in the database—data file, data block, and position of the data inside the block. It, however, does not consist of the object identifier, whereas it would not provide any additional information relevant for the evaluation process, obtaining result sets. In our case, the updated ROWID value consists of just 6 bytes (data—1.5 B, data block—2.5 B and 2 B are used for storing block position).

Evaluation and performance reflection has been done in the same environment and data, as already described. In this part, we deal with three models and their extensions to identify historical data and versions. For the purposes of this paper, we compare the strategies expressed by the processing time. Performance can be compared to the referential model managed by the archiving mode of the database (model M0). As shown in Figure 14, model M6 based on removing historical data from the archive repository does not bring any benefit expressed by the processing time, whereas particular data files are sequentially numbered, and historical data are automatically removed from the evaluation process. Naturally, it would be significantly better in the structure size spectrum, which will be evaluated in the second phase. Extension of the concept modeled in M6 is expressed by the M7, which adds a layer with temporal pointers. Thanks to that, temporality, versioning, and corrections can be easily identified. The pointer layer is secured by the three models. Model M7a uses varray as the structure, while model M7b is characterized by the nested table. The disadvantage of such a structure is just the indexing strategy, which cannot be used, whereas nested table structures cannot be indexed, forcing the system to scan and evaluate data sequentially. The best solution for the model M7 proposes model M7c, where the pointer layer is extracted into a separate table covered by the indexes to ensure searching effectivity.

Model *M8* is based on the versioning, either embedded (M8a) or modeled by the pointers (M8b).

Performance results in terms of processing time expressed by the percentage improvement are in Figure 14. The reference model M0 and model M6 reach the same results, whereas the model M6 just adds the module for the identification of expired archive log data files, which are not, however, scanned during the processing and historical data identification, whereas each such file consists of the time positional data in the header. Moreover, individual log files are time sorted. Improvement of 8% brings model M7a, in which the pointer layer is modeled by the varray. In such a case, the appropriate size for the varray must be defined, whereas it cannot be extended higher than the upper limit (attribute *limit* of the varray structure defined during its creation). M7b uses a more generic pointer layer modeled by the nested table, reflected by the 12% improvement of the processing time. The strength of the index approach can be identified in model M7c, where the performance improvement is 21%. When dealing with the versioning, the performance benefit ranges from 26% (model M8a) up to 30% (M8b). The performance results are shown in Figure 14.

The second aspect to highlight is the size of the whole structure. Model M6 does not bring any additional size demands, whereas it just launches the process to identify log files in the archive repository, which can be deleted. Models in the M7 category add a new pointer layer. They require additional costs from 2% (M7b) up to 4% (M7a). The reason is based on the fixed size for varray used in M2a. When dealing with a separate table for the pointers (M7c), size demands are higher—3%, whereas it uses an additional index structure in comparison with M7a and M7b.

Model category M8 highlights the temporal versioning of the transaction aspect. Embedding change vectors directly within the system data requires just 74% of the processing time (M8a), which reflects the improvement from 5% (M7c) up to 18% (M7a). Grouping data and change vectors into the inseparable system is just the performance limitation, whereas the structure is fixed and too strict. As can be seen in Figure 14, a reference to the change vector structure by using pointers should be highlighted. It is represented by the model M8b. By using such a solution, the processing time is lowered and expresses just 70%, which provides a 4% benefit in comparison with the embedded system (M8a). Comparing the solution with other systems, improvement ranges from 9% up to 30%.

### 5.3. Shared Pool

When dealing with the performance, data access, and processing tuning, another relevant memory structure is the shared pool and specifically its subpart library cache.

The shared pool is the most complex memory structure covering dozens of elements. From the performance point of view, the most relevant types are library cache, data dictionary cache, PL/SQL area, and SQL and PL/SQL result caches. In this paper, we have focused on the library cache, which stores the execution plans of the recently executed queries (Figure 15). It works like the linked list based on the relevance and usage; thus, the most historical and least relevant data are removed, if new free space is necessary to be allocated. It holds the SQL code in its parsed form identified by the hash value of the processed code. Therefore, it is easy to identify whether such code (or its similar version) has been already evaluated, and the execution plan is available. If so, it is not necessary to parse it one more time, and the processed plan is taught and used. As described, when dealing with the data, they can be either obtained by using the index or by sequential scanning of the whole table. Our proposed solution brings the extension that index is used just as the data locator layer to remove the necessity of scanning each data block, even irrelevant or empty. The database optimizer chooses the best suitable way, based on the statistics and system parameters. The particular execution plan is then stored in the library cache. From time to time, the system evaluates whether a particular plan is still effective, reflecting the execution plan baselines [29]. Data access using the master index is stored in the execution plan list, as well, but influenced by these rules:If the new index is defined for the table, each stored execution plan reflecting the particular table is recalculated and denoted as expired, whereas the new index structure can be more relevant for the query.The newly developed index does not need to be suitable for the query itself, resulting in using the TAF method. It can be relevant for using it as the master index, as well.If the existing index is dropped (mostly as the result of the monitoring usage [24]), particular master approaches for the table are checked, whether they used such index as master or not. If so, these execution plans are marked as invalid.

Master index selection can be either static or dynamic. If the static approach is used, the optimizer uses the preselected master index, based on the stored execution plan in the library cache. It, however, does not need to be optimal in the current environment, whereas the current selection takes the emphasis of the index loading costs, as well. Naturally, later, there can be different situations and conditions of the memory allocation, index loading, free buffer cache blocks, etc. Therefore, we introduce the dynamic mode of the master index. During the process of data retrieval, database optimizer looks for already prepared execution plans. If they are present in the library cache, three situations can occur, reflecting the data access:Table access full (TAF)—in this case, no suitable index can be denoted as master, or the master index principle is disabled during the previous retrieval.Index scan—index access is used for accessing data rows.Flower index approach (FIA)—The master index is used to locate data inside the data files using block granularity.

For the optimization aimed in this paper, the main property is the difference between TAF and FIA approaches. If any index covering all data tuples is defined, the FIA approach can be present and enabled. Moreover, it can automatically reflect any change—adding or restructuring an existing index list. Thus, master index selection can be even improved. FIA represents the flower index approach as the mechanism for dealing with and selecting the master index. Dynamic access allows you to change the access index for the FIA. The master index is the best appropriate one based on the execution timing—the time at which the plan is created and stored. However, the environment and technical details may be changed later, so the system can re-evaluate the suitability of the index to mark the master definition with the emphasis on the current processing costs (loading into the memory, size, fragmentation, etc.). Data flow UML diagram of the data and evaluation flow is shown in Figure 15. In this case, the processing is dynamic; therefore, the master index can be changed during the evaluation process.

### 5.4. Data Dictionary Cache

The data dictionary is a core part of the metadata of the database system and objects inside. It stores recently used object definitions, like descriptions of the tables, index characteristics, users, etc. Keeping such data in memory brings performance benefits. Our solution shifts the importance and definition to the next level. It divides it into two categories based on the structure and data they hold. The standard existing approach is based on the circular structure, where historical data are consecutively replaced by newer versions of the other metadata. Such an approach is, however, not sufficient for the transaction log data management. As explained in the previous section, in our case, log data from the transactions reflected by the linked list of the changes are stored in the data dictionary. During the processing, they are loaded into the memory but are not replaced, if there is no available memory for the structure inside the dictionary cache. Instead, this memory structure is dynamically expanded to cover all requirements. We emphasize efficiency, so we do not store log data themselves, but merely descriptive pointer data that sort individual changes and refer to the transactions.

Robustness, security, and independence of the database systems are characterized by the total abstraction of the logical definition covered by the instance and background processes and the database storage. From a logical perspective, interconnection is covered by the segments, which are covered by the definition of the object itself, like the table, index, etc. Segments are physically stored in the data files. Abstraction itself is delimited by the tablespaces. Definitions and relationships are maintained in the data dictionary.

There are three structures that are part of the database system core—controlfile, consisting of data repository location and initialization parameters. It is used during the process of database mounting and opening. The second category covers online redo log files containing change vectors copied from the memory structure into the physical storage. This structure is covered by several files, which are rewritten in a circular manner if they do not hold current transaction data. It is operated by the log writer background process and accessible by the archiver, as well. The archiver background process is important if the online logs are copied into the archive repository. The whole process is guarded by the archiver.

The third required file type is delimited by the data files. At a minimum, two data files must be present at the creation time covering SYSTEM and SYSAUX tablespace. Data files are the repository for the permanent data. They are the physical structures associated with the tablespaces. Inside them, individual segments and extents are present in the block group shape. A segment is a storage structure of the data. Each tablespace consists of several data files, characterizing the segments and data extents as a set of blocks.

## 6. Tablespace, Segment, Data Block

Data are physically stored in the data files located in the database. These files are formed in as the list of individual blocks, whose characteristics are defined by the tablespace. A tablespace is a logical unit, abstracting data from the instance and storage. It defines block size, which is an important part of the processing and loading, whereas I/O operations are the most demanding step, influencing global performance.

### 6.1. Tablespace

Physical storage for the data logically formed as segments and extents is in data files. The tablespace is the entity defined in the relational paradigm, which abstracts logical and physical storage levels. Thus, each user addresses just the logical structures regardless of the physical implementation. Thanks to that, storage structures and their management can be changed without the necessity to compile applications or implement new rules and techniques. Figure 16 shows the storage model used in the Oracle database, which is used for the evaluation; however, the principles are similar to any relational database system, whereas its paradigm is always implemented.

The cardinality of the relationship between database block and tablespace is one-to-many, whereas one tablespace can cover many segments, and these segments are defined by more than one database block as the result of the allocation—blocks are not associated with the segment in an individual manner, but as the sets named as extents. As evident, one segment can be spread across multiple data files, which was not, however, possible in the past [1].

### 6.2. Segment

The segment represents any database object that stores data, so it requires space allocation in a tablespace. The typical segment is a table, index, but there are transaction undo segments, as well. The segment can be just part of one tablespace, but it can spread across many data files, too.

### 6.3. Data Block

A data block is a main unit of the I/O operation and loading into the memory. It is not possible to transfer a smaller piece of data. Data blocks are consecutively numbered and are part of the segment and extent. The size of the data block is fixed across the whole tablespace. Many times, the database uses the same block size in the whole spectrum, and, by default, it represents 8 KB. A row itself can be represented only a couple of hundred bytes; thus, many rows can be present in one database block. To get the value of the state—the image of the row—it is necessary to load the data into the memory buffer cache in the form of the block. As evident, the block is the core granularity for the processing; therefore, its size is crucial for the evaluation and performance itself.

The size of the database block can range from 2 KB up to 32 KB in specific releases controlled by the parameter DB_BLOCK_SIZE. Our proposed solution is based on using the index as the data block locator; therefore, it is inevitable to discuss and experiment with the block size, parameters, and performance impacts.

The following block shows the syntax script for the tablespace definition. Each of them is delimited at least by the unique name and data file associated with it. Size of the block is defined in the fifth line preceded by the extent size management,

Create tablespace mk_tablespace

Datafile ‘/home/kvet/tablespace/mk_tablespace.dbf’

Size 500 M

Extent management local uniform size 1 M

Blocksize 8 K;

The selected block size must be compatible with the storage inside the memory. There must be a reserved capacity of a block matrix with a particular block size secured by the DB_CACHE_SIZE parameter of the database instance.

Alter system set db_16k_cache_size =64 M;

Current parameter values can be obtained by querying v$parameter view:

Select name, value

From v$parameter

Where upper(name) like ‘%_CACHE_SIZE%’;

Block size, structures, and parameters can have a significant impact on the performance, whereas the block must be loaded into the memory before the processing itself. In this section, performance analysis is present with an emphasis on the block size and loading technology. In the first phase, we deal with the block capacity and consecutive fragmentation compared to the default block size—8 KB. Block size and block assignment are defined by the tablespace associated with the particular table. To change the capacity of the block, it is necessary to associate the memory structure to allocate the block matrix structure to be available for the block loading. After executing it and accepted by the database system, tablespace with the 16 KB block size can be created. Loading and management are secured by the same background processes as the default principle; however, the memory structure and location are different. Without the association of the memory structure for the defined block size, it is even not possible to create the tablespace itself.

Current database environment parameters are available by querying the v$parameter performance view. In the following section, performance comparison and impacts on the block size are reflected.

All of the experiments are provided 10 times, the shown results express the average values, whereas several impacts, like external processes outside the database, OS maintenance processes, network communication, etc., can influence results; therefore, such impacts are limited, and the complex significance is lowered.

In the first phase, the size of the whole structure is evaluated. The same data and operations are performed on all of the systems distinguishable by the block size definition. Figure 17 shows the number of blocks in the percentage. The referential model is 8 KB as the default setting for the database systems. Environmental characteristics are the same as described in the previous section of this paper.

The characteristics are nearly linear. If the basic block size is larger, then, naturally, fewer blocks are necessary to be assigned by the system. The 2 KB block size structure requires 430.7% of blocks, in comparison with the referential model (8 KB). Vice versa, 32 KB block structure requires only 23.1%, but the capacity is quadrupled. Thus, after the transformation to the default environment settings (8 KB), the system saves approximately 7.6% of the block count capacity. Blocks are, however, not allocated separately but are formed as the set to the extent. The principle of the results expressed in Figure 16 is based on the assumption that the block and extent itself is the same expressed by the 1:1 mapping. One extent is in that case bounded by just one block, which is not, however, appropriate from the performance point of view, in case of changing data, inserting new rows, or updating existing tuples. The allocation process namely is resource consuming and requires server components coordination and their usage. Therefore, blocks are not created separately but by using extents. Their size and allocation principles are based on the configuration of the tablespace. In our case, the uniform size is used. Correlation between block and extent capacity can be found in [1,2,3]. Figure 18 shows the number of extents calibrated to the default block size, 8 KB, expressing 100%.

Based on the reached results, it is clear that the changing capacity of the block does not provide a significant impact in terms of the capacity and size demands for the database storage. In our case, each extent consists of 12 blocks. Thus, if the block size is 32 KB, the total required capacity is even higher in comparison with the 8 KB data block. Total consumption is about 20% higher. Vice versa, when the block size is lowered, it requires increased costs, as well. The value of the increase is approximately 9%, in comparison with 8 KB block size capacity.

To conclude the performance study with emphasis on the block size and the number of allocated extents, it can be solidly stated that restructuring the system on the block layer and changing the core block size do not provide the benefit. Notice, however, that the block allocation is done just as the result of the insert or update statement as the result of sensor data values flow.

Another significant impact to be highlighted is associated with the data retrieval for the result set composition, analytics, reporting, etc. Therefore, in the second part of the performance study of the block size, the performance relationship is evaluated and expressed by the loading process into the memory. To provide data and create result sets, it is necessary to load relevant data into the memory buffer cache, forcing the system to load particular data blocks from the database into the instance memory.

The data retrieval process consists of multiple steps to be done. Index management is selected by the optimizer during the parsing and execution plan composition process. The experiment section regarding data retrieval can be divided into multiple phases. In the first phase, index definition and suitability aspects are taken into account. The second phase includes the analysis of the block size and reflection of the index processing. The third phase is associated with the physical loading process—I/O and data transfer.

#### 6.3.1. Data Retrieval–1 Phase-Index

For the evaluation, five scenarios are evaluated. Performance originates from the sensor data flow for the evaluation. The first scenario is based on the situation, where no index is present in the system. In that case, the table access full (TAF) access method must be used. The second scenario is based on the index, which contains all data for the processing and result set composition. Thus, the physical database does not need to be addressed. Therefore, the second scenario does not have any relationship to the physical database storage and data block size, whereas the database itself is not contacted at all. The third scenario uses an index for the evaluation; however, one data attribute is not part of it; therefore, database block loading is present during the processing to obtain missing attribute values. There are two access methods—index range scan followed by the table access and index ROWID obtained by the index. On the leaf layer of the index, ROWID values as the direct locators for the data are there. The fourth scenario is based on the index; however, the order of the attributes is not suitable for the processing; therefore, the whole index must be scanned. More about the order of the attributes inside the index and performance impacts can be found in [23,24]. The last scenario uses our proposed FIA approach. Table 1 shows the results. In the table, processing time and costs are evaluated. The database query costs parameter expresses the result of the function, reflecting all resource demands—processing time, CPU, I/O, communication, instance processes, etc. When dealing with the optimal index for the query, total processing costs can be lowered from 1850 to 512, which reflects the improvement using 72.3%. If the index scan must be followed by block loading, the total processing costs are reflected as 1379 (25.5%). A comparison of the TAF method with the FIA approach recommends the master index, where the costs are lowered from 1850 to 705, corresponding to 61.9% improvement. In this case, the processing time is lowered from 72 to 29 s (59.7% improvement).

When dealing with the data access, a relevant approach for the comparison and evaluation is the index-organized table (IOT) approach, as well. In that case, data are shaped and formed based on the primary key, not heap. Table 2, Table 3 and Table 4 show the reached results. Three models are compared. The original referenced type is TAF, in which no index is presently forcing the system to perform sequential block scanning [1]. Our proposed approach cannot be used at all. The second solution deals with IOT, where several indexes are present. The limitation of this approach is based on the usability and suitability of the index. If no suitable index is defined, instead of TAF scanning, the primary index is always used, regardless of the costs of the index loading, processing, and condition evaluation. Our proposed solution removes such limitations. Therefore, if no suitable index is present in the system, the master index approach (FIA) selects the best one (from the secondary index list) appropriate based on the current conditions.

The first part of the evaluation experiment is based on using only one index in the system defined by the primary key. In this case, the processing costs of the referential no-index environment are 1850, and processing requires 72 s. Shifting the management into IOT with just one primary index (no secondary indexes are present) lowers the total costs to the value 703, requiring 29 s. The solution defined in this paper (FIA master approach) reaches almost the same results, i.e., 705 of the processing costs and 29 s. The difference between costs is less than 0.3%. Thus, there is no relevant difference between IOT and FIA approaches, and the performance impact is minimal. On the other hand, management compared to TAF gets strong improvement, even up to 72% for the costs, and 60% for the time evaluation. Table 2 shows the results based on the environment, where just the primary key defines the index set.

In the previous environment described and in Table 2, no secondary indexes are present; thus, the master index cannot select from the index set, whereas its volume is 1. In the following part, we deal with several index structures. Thus, the previous approach is extended by the secondary indexes. We use each suitable attribute as an index definition. Suitability of the index is delimited by the NOT NULL constraint. Whereas NULL values are not stored and referenced in the index, the particular index is not meaningful for the master index evaluation in the FIA method. As you can see from the reached results (Table 3), IOT results are not changed at all. Whereas no suitable index for the query, the primary index defined by the structure is used, regardless of the existence of the secondary indexes. In FIA approach, master selection optimizer automatically chooses the best appropriate index, based on two conditions:costs of the loading index into the memory,size of the whole index.

As you can see from the results, costs can be lowered to 423, whereas composite primary key index can be separated into individual attributes, which are naturally smaller in comparison with the whole set. In the experiment, we use the assumption that the secondary index size is 50% of the primary key index. The total costs reflect a 40% improvement. Based on further experiments, we come to the conclusion that the ratio between the size and costs is even almost linear. Processing time is lowered from 29 s (IOT) to 17 s for FIA. It represents a 41% improvement. Table 3 shows the reached results. Based on this experiment, it is clear that secondary indexes play an inseparable role for the optimization, whereas IOT does not select them if the order of the attributes inside is not relevant for the query. Thus, secondary indexes in IOT are not used as the data locators at all.

The third part evaluates the costs for an already partially loaded and processed index. In this section, we assume that one half of the index is already present in the memory buffer cache; therefore, only missing part must be loaded before the evaluation and data location itself. Such an already loaded index is part of the secondary index set. As evident from the results shown in Table 4, our proposed FIA approach benefits from the current environmental situation, whereas IOT does not reflect it at all. Total costs of the TAF (1850) are lowered to the value 410 for FIA, which represents 77.8% improvement. Comparing FIA and IOT on the costs layer, we come to the conclusion that FIA requires only 42% of the costs. Reflecting the processing time, we reach the saving of 12 s, representing 41.3%. To conclude this section, it is clear that if secondary indexes are present in the system, the master index automatically evaluates them, and its decision is based on the current situation in the system with an emphasis on the index size, costs of the loading, and block size, as well. Usually, index master selection is done automatically, and if the developer would be able to select the optimal solution on his own, the processing benefit would be approximately 6%. The results are in Table 4.

#### 6.3.2. Data Retrieval–2 Phase-Index and Block Size

In the previous part, the index range scan method followed by data access using obtained ROWID values lasted 62 s in total and required 1379 costs. In this section, block size and performance impacts are highlighted. For the purposes of the analysis, 2, 4, 8, 16, and 32 KB block size is used. To propose relevant results, the environment properties and characteristics are the same. The size of the memory structure for the loading is unlimited and is set to higher values than the index and data themselves. No other transaction is present in the system, ensuring no data transfer from the memory to disc as the result of full memory storage can be present.

Obtained results are stored in Table 5. As you can see, the size of the index does not differ significantly. In principle, the index still refers to the same data, which are, however, formed in the different block structures. Costs of the processing include also the loading costs, waiting for I/O and loading itself, and, therefore, the shift can be recognized. Table 5 shows the processing costs and total processing time. As can be seen, there is no significant difference in terms of performance reflected even by the processing costs and time, as well. For the index costs, the improvement is approximately 2%, and for the data access itself, the improvement is 10% (referential model is 8 KB—default approach).

Based on the reached results presented in Table 5, costs vary from 388 up to 400, which expresses the difference as a 3% slowdown. When dealing with a 2 KB data block, data access costs have the lowest value (880). With the increase of the block size, costs are rising due to I/O operation and loading necessity. Namely, the data block is the core granularity; thus, no smaller data portion can be loaded. An important aspect is a ratio between the row size and the total block capacity. A 4 KB block brings a 5.9% increase in costs. A similar increase is identified (5.5%) in the transfer from 4 KB to 8 KB. A 6.3% processing cost increase is defined when shifting the granularity from the 8 KB to 16 KB. To conclude the evaluation of the processing costs, it is clear that doubling the block size brings about a 5%–6% increase in costs caused by the loading process and ratio between the block and row size. Processing costs increase vary from 1 up to 3 s, which reflects a 1 up to 4% increase.

Different situation, however, occurs, if the memory size structure is not efficient and transfer between database and memory is bi-directional. In that case, the results are following (Table 6):

The results shown in Table 5 are based on the fact that the memory for the processing is suitable; thus, all the defined data can be located in the memory. In this case, the dynamic transfer must be present to get the buffer cache available for the block processing. When using the 2 KB block size, the total costs are 1275. When using 4 KB block size, costs are 1386, with an increase of 111 in the costs (expressing 8.7% increase). In comparison with unlimited memory size, demands are increased using 2.8%. By using the default block size (8 KB), the costs are almost 1615. The increase in costs is 16.5%. Comparison with unlimited memory delimits increase of 11%. Another greater block size definition has almost the same results. The reason is based on the block capacity, which can hold a significant amount of data rows. In that case, the transfer from the memory into the database does not play any role. The increase in costs is lower than 1%. The results are shown in Table 6.

#### 6.3.3. Data Retrieval–3 Phase-Physical Loading Process

The result of the previous phase was enhanced by the block size and performance implications. In this part, we split the whole process of the data access inside the database into these steps:Identification of the corresponding data blocks (operation 1).Request for data transfer operated by instance processes and supervised by the operating system (operation 2).Waiting for I/O operation (operation 3).Loading (operation 4).

Table 7 shows the results of the processing and loading expressed by individual operation costs. As you can see from the results, operation 1 requires approximately 8% of the total processing costs (from 7.8% for 2 KB block up to 8.5% for the 16 and 32 KB blocks), operation 2 is not demanding and requires approximately 2% (from 1.8% for 2 KB block up to 2.5% for the 32 KB block). The core part is delimited by the operation 3 and operation 4. Proportion regarding operations 3 and 4 differs based on the block size definition. For the 8 KB block size, it reflects 30% for operation 3 and 60% for operation 4. If the block size is higher, the impact on the loading (operation 4) is increasing, and the importance of operation 3 is lowered. If the block size is 32 KB, the proportion of the operation costs are 8.5/2.5/11.5/77.4 %.

The graphical representation is shown in Figure 19.

## 7. Discussion

With the growth of the data amount to be processed, it is necessary to optimize approaches, architecture, and access methods to ensure a reliable, robust, and powerful solution. Many times, the data to be processed originate from the sensor environment, characterized by the significant data stream, as well as the data block fragmentation, which is the result of transferring historical data into separate repositories or just as the result of versioning and update operations. Performance depends on many factors and parameter settings for the specified system characteristics and application domain. The core part of the optimization is related to the index usage. The first part of the paper, therefore, deals with the index access techniques. If the appropriate index for the query is defined, the index unique scan or index range scan method is used based on the condition defined in the Where clause of the select statement. If the attribute order reflecting index and query itself is not suitable, two choices are available—particular data are part of the index, but the attribute order is not correct. In that case, the full index scan method category can be used. Data inside the index are namely smaller and possibly already located in the memory; thus, the loading process can be omitted. The second approach is more complicated and is characterized by the environment, where no suitable index is defined for the query, forcing the system to scan all data blocks sequentially. Physically, it means that the whole block must be loaded into the memory buffer cache, where it is parsed and evaluated by extracting data rows. The limitation of such an approach is just the performance. The system cannot delimit usage of the block and its fullness, consequencing in the necessity to load even empty and partially empty blocks. Therefore, index usage is significant for query performance. On the other hand, the index definition requires many system resources during destructive DML operations (insert, update, and delete), whereas the change must be applied in all indexes before the approving transaction.

Our proposed flower index approach and selection of the master index create a new layer between the index and all block scanning methods by interconnecting them. In the proposed case, the index is used just as the data locator. It is enhanced by the master index definition to lower the processing costs to the minimal possible value. A dynamic approach is very convenient, whereas it reflects the current state of the database and the instance and prefers the usage of such index as the master one, whose costs of the loading is minimal. Thus, the selection and evaluation are based on the indexes, which are already at least partially loaded in the memory. Based on the experiments, we can perceive a significant improvement in comparison with the TAF method, even up *to* 80%, based on the fragmentation and proportion of required data to the whole data set (expressed in percentage). Thus, it is not necessary to create a sophisticated suitable index for each query, but the whole performance is ensured.

Sensor data layer management can be characterized by the strong data flow and value evolution over time, enhanced by the detection of the failures and significant changes. At the same time, historical data lose their validity and significance in the system and are moved to other repositories or possibly completely removed. As a consequence, data fragmentation is present on the physical block layer. Therefore, our proposed FIA approach defines substantial progress in terms of performance, whereas non-relevant data blocks are not scanned at all.

Another aspect is reflected by the reliability and data correctness. Therefore, it is necessary to cover the whole data management process, from the data loading into the system up to retrieval. We take the emphasis on the physical data block architecture and size. For those purposes, we evaluate the impact of block size on the performance strategy. The whole experiment evaluation is divided into several steps. The aim is to state block size and implications if the block size granularity is changed. Based on the performed computational and execution study, we come to the conclusion that data access corresponds to the block size, not more than 10% for the processing costs. Thus, the difference in performance costs in terms of the size of the allocated block is a maximum of 10%. On the other hand, proportional changes for individual operations can be felt. The identification of the relevant blocks from the index does not have any impact on the performance if the block size is changed. The reason is based on the pointers, whose amount is not changed, just the location of the data is divided based on the block size and free available capacity. A similar situation occurs when requesting data access, and there is no relevant difference. Proportional characteristics changes can be, however, recorded during the waiting process and data transfer from the database to the memory. If the block size is just 2 KB, data loading requires 740 costs, which represents almost 58% of the whole processing. In comparison with default block size (8 KB), where the loading spends 870 costs expressed as 60%, there is only a slight difference. However, if the block size is expanded to 16 KB, data transfer represents 912 costs, which expresses 68% of the processing costs. For the data block size of 32 KB, it even represents 1034 costs (79%). Therefore, it is clear that such problematics is highly relevant with the aim to lower I/O operations and data transfer from the database to the memory and vice versa. Naturally, available memory capacity has an important impact, as well, but it is technically impossible to extend the hardware capacity to an unlimited manner, especially by emphasizing the fact that the amount of data is growing steadily.

To conclude this paper, based on the computational study, it is clear that the master index is a robust and suitable solution, replacing the necessity of sequential scanning of all blocks associated with the table covered by individual extents. If the index covering all data rows is present and denoted as master, it is used as the pointer layer for the blocks holding relevant data. The extension is based on the block index, in which the leaf layer does not consist of the ROWID values themselves, but identifiers to the block granularity are used instead. Thanks to that, significant performance improvement is reached. The system is based on the relational paradigm, which can manage data from any data source, completely covering the sensor data orientation.

In the future, the management of the synchronization groups as the result of spatial databases will be focused on. Sensors can be grouped together by one communication channel, which sends results in the batches. Thus, in that case, it will benefit if such data portions are always processed as a transaction.

## Figures and Tables

**Figure 1 sensors-20-01848-f001:**
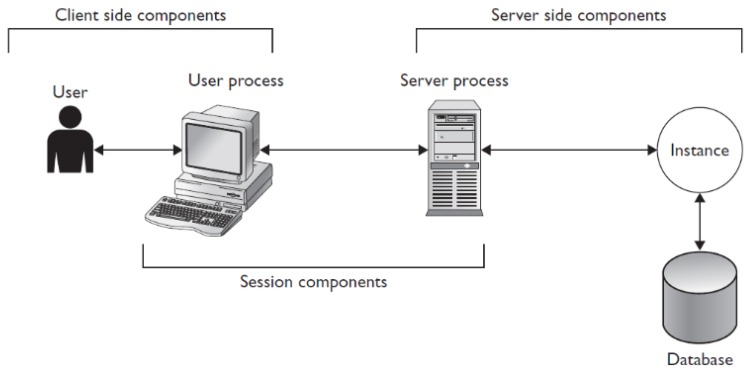
Client-server architecture [1].

**Figure 2 sensors-20-01848-f002:**
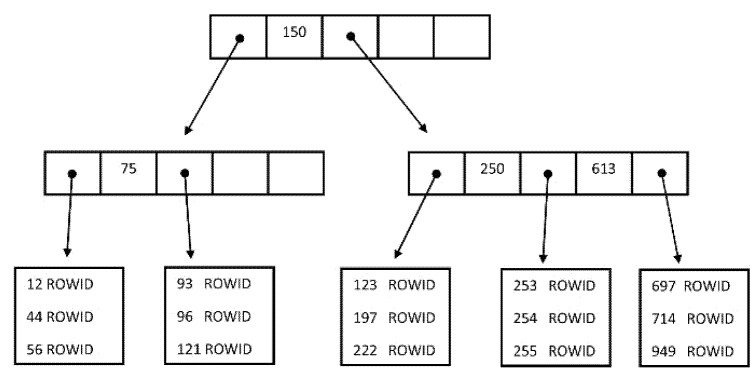
B-tree structure containing ROWID pointers in the leaf layer.

**Figure 3 sensors-20-01848-f003:**
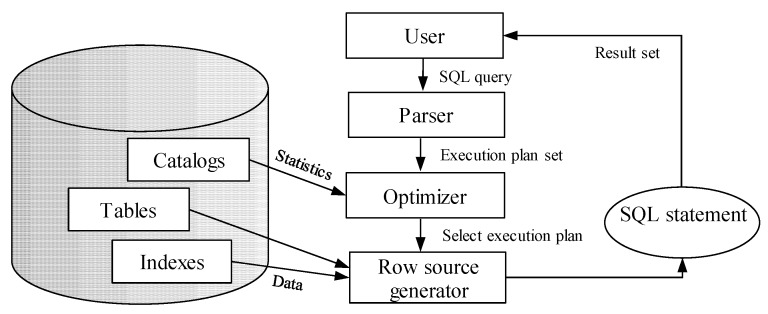
Query evaluation [1].

**Figure 4 sensors-20-01848-f004:**
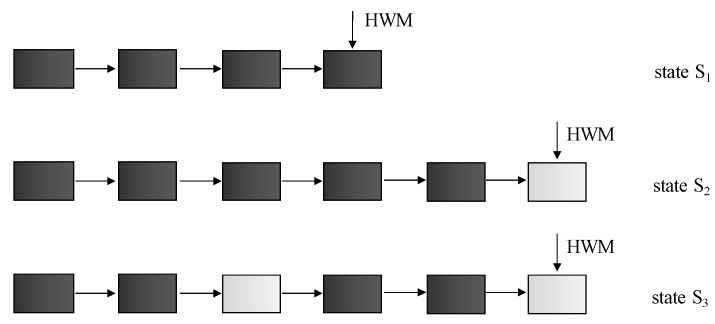
Block fullness. HWM, high water mark.

**Figure 5 sensors-20-01848-f005:**
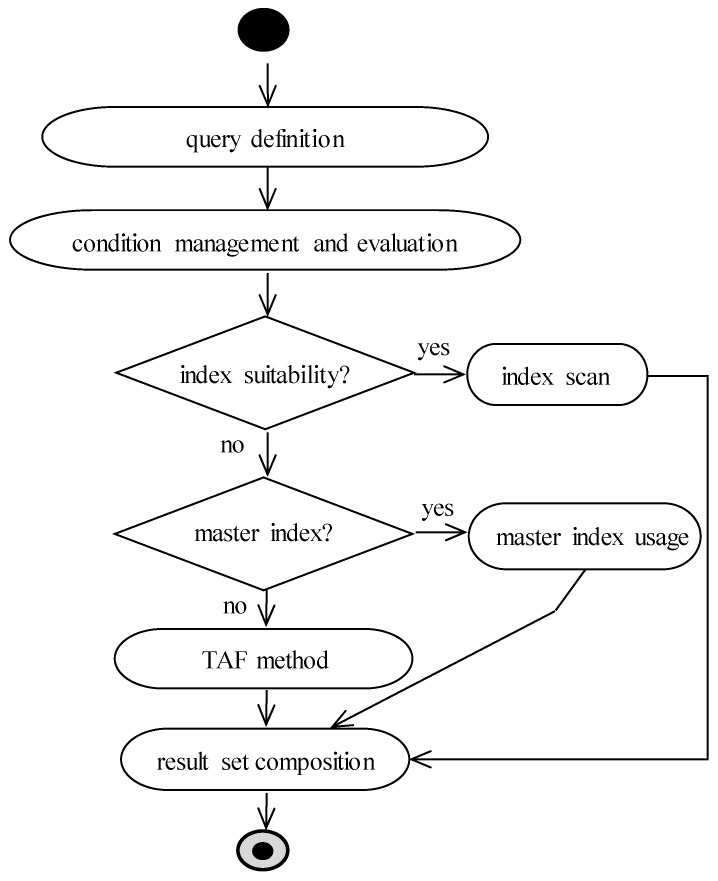
Data flow—Index access selection.

**Figure 6 sensors-20-01848-f006:**
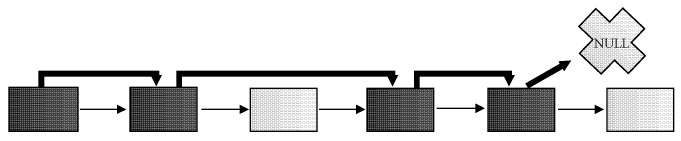
Model 2.

**Figure 7 sensors-20-01848-f007:**
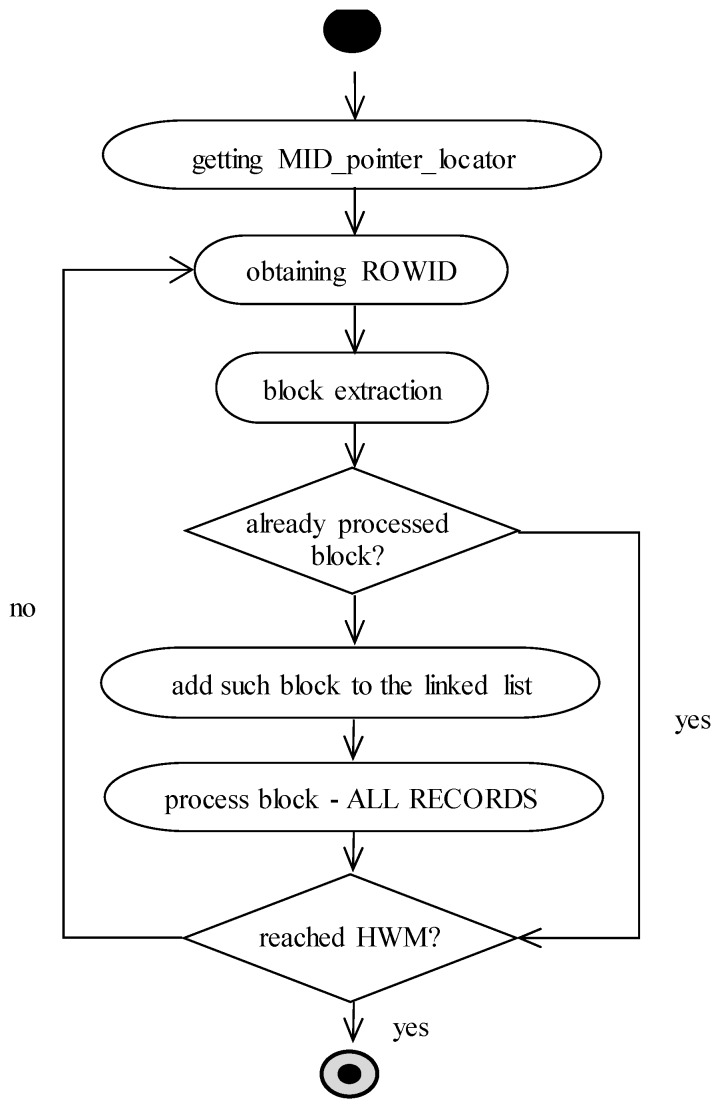
MID_pointer_locator and consecutive data management. MID, master index definition.

**Figure 8 sensors-20-01848-f008:**
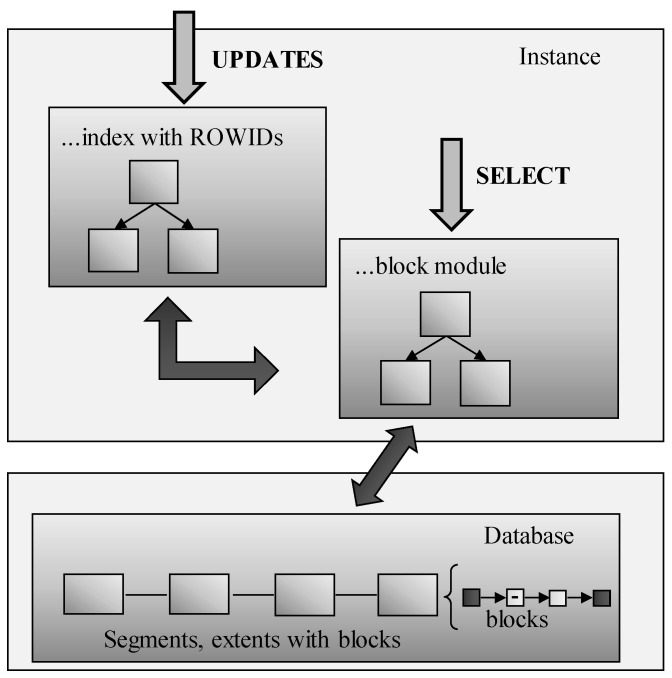
The architecture of the solution—block management.

**Figure 9 sensors-20-01848-f009:**
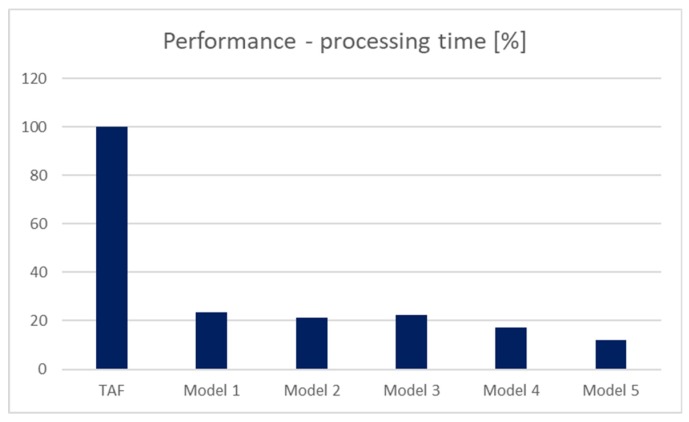
Performance results—processing time (%).

**Figure 10 sensors-20-01848-f010:**
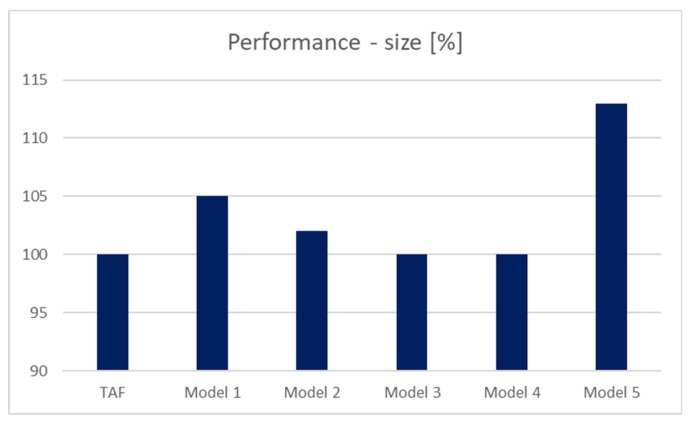
Performance results—structure size demands (%).

**Figure 11 sensors-20-01848-f011:**
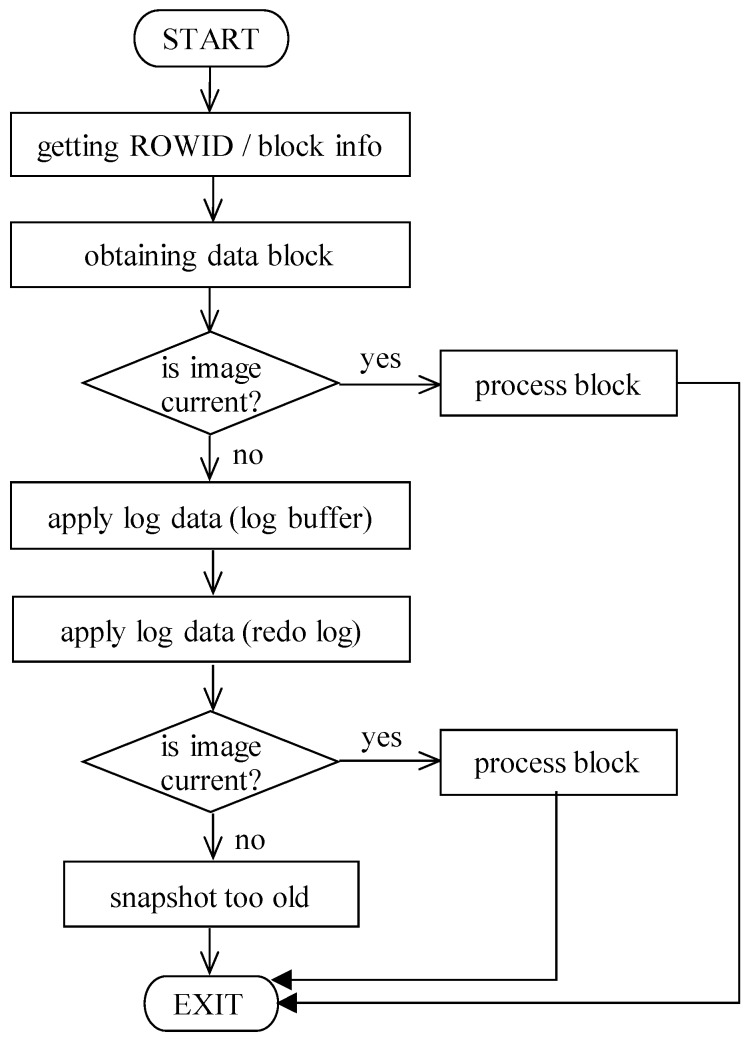
Process of block data reconstruction.

**Figure 12 sensors-20-01848-f012:**
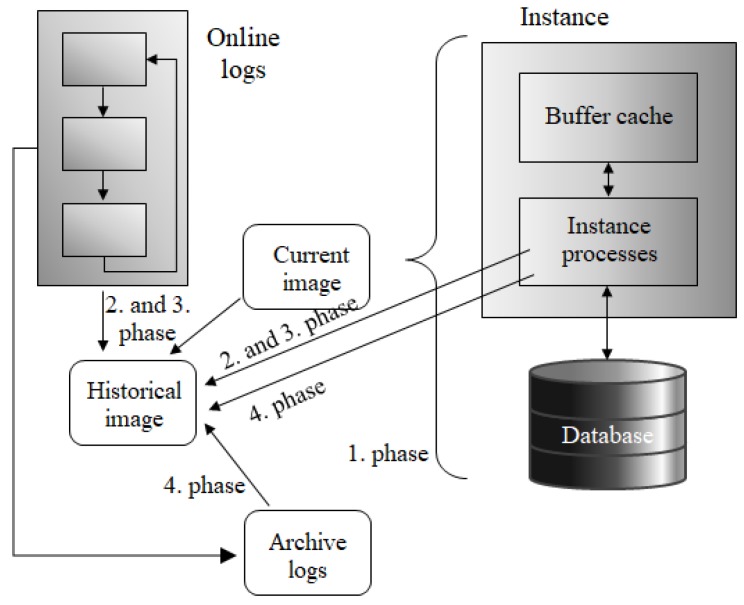
Archive log management.

**Figure 13 sensors-20-01848-f013:**
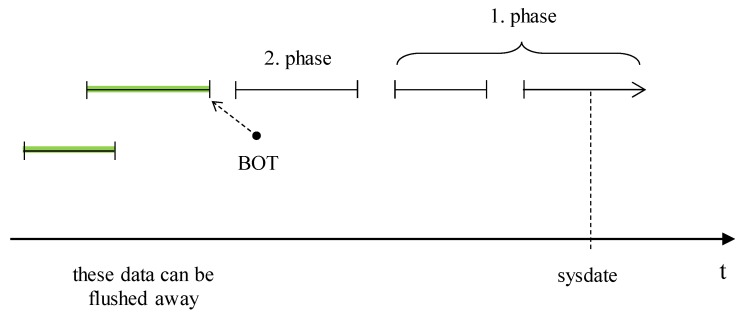
Expired data identification.

**Figure 14 sensors-20-01848-f014:**
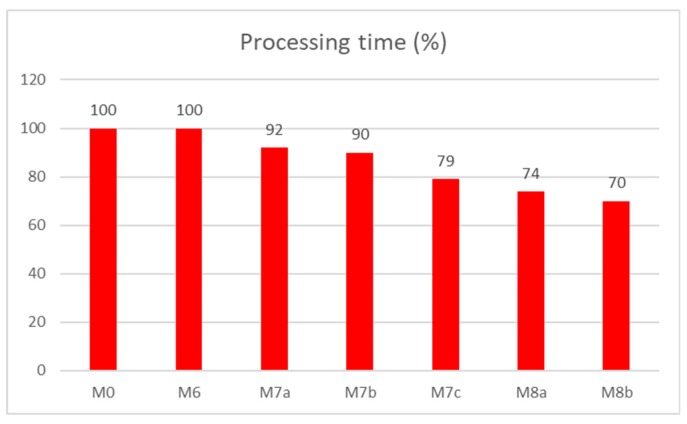
Processing time—pointer layer.

**Figure 15 sensors-20-01848-f015:**
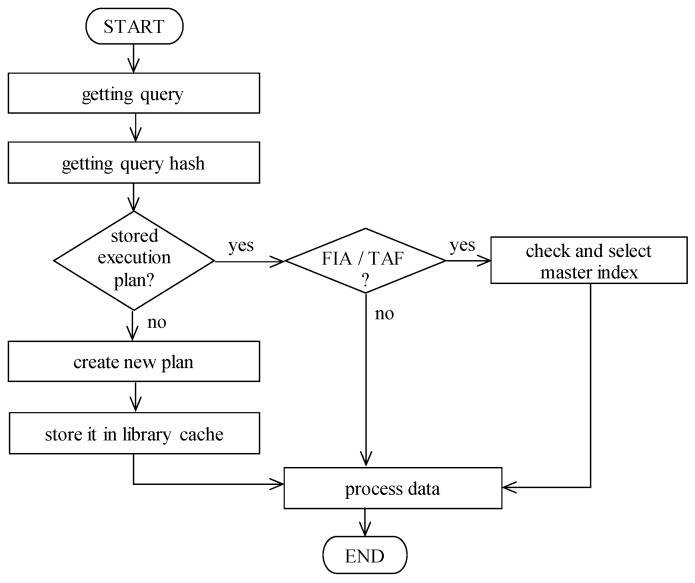
Managing and evaluating execution plans. FIA, flowering index approach; TAF, table access full.

**Figure 16 sensors-20-01848-f016:**
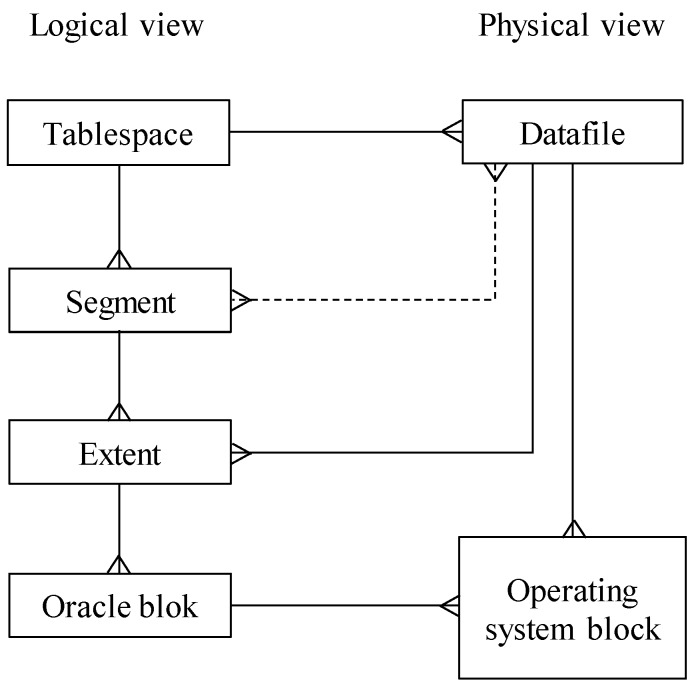
Database storage model.

**Figure 17 sensors-20-01848-f017:**
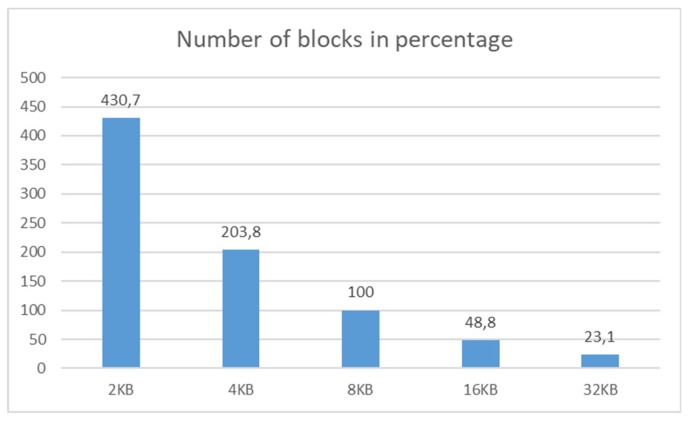
Results—block count expressed in percentage.

**Figure 18 sensors-20-01848-f018:**
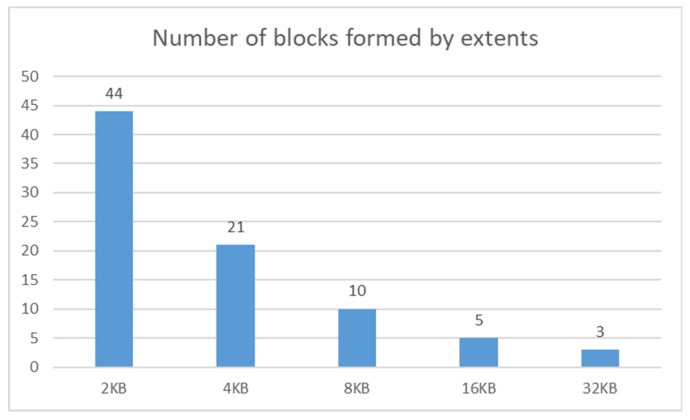
Results—number of blocks formed by extents.

**Figure 19 sensors-20-01848-f019:**
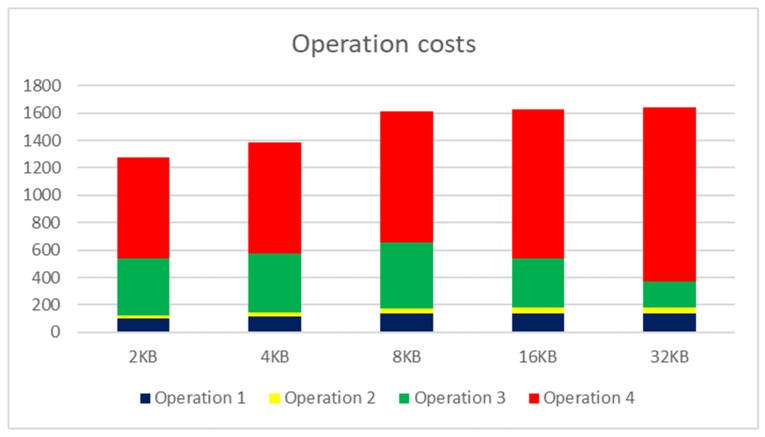
Operation costs.

**Table 1 sensors-20-01848-t001:** Index processing—costs and time demand.

	No Index	Suitable Index, All Attribute Values are Present	Suitable Index, One Attribute is Missing		Index, Whose Order of Attributes is not Suitable	Flower Index Approach
Access Method	Table Access Full	Index Range Scan	Index Range Scan	Table Access by Index ROWID	Index Fast Full Scan	Flower Index Approach
Costs	1850	512	396	983	881	705
Time (hh:mi:ss)	00:01:12	00:00:25	00:00:18	00:00:44	00:00:37	00:00:29

**Table 2 sensors-20-01848-t002:** Index processing—costs and time demand in one index environment.

	No Index	Index Organized Table	FIA
Access Method	Table Access Full	Index Range Scan	Flower Index Approach
Costs	1850	703	705
Time (hh:mi:ss)	00:01:12	00:00:29	00:00:29

**Table 3 sensors-20-01848-t003:** Index processing—costs and time demand—the presence of the secondary indexes.

	No Index	Index Organized Table	FIA
Access Method	Table Access Full	Index Range Scan	Flower Index Approach
Costs	1850	703	423
Time (hh:mi:ss)	00:01:12	00:00:29	00:00:18

**Table 4 sensors-20-01848-t004:** Index processing—costs and time demand—index loading.

	No Index	Index Organized Table	FIA
Access Method	Table Access Full	Index Range Scan	Flower Index Approach
Costs	1850	703	410
Time (hh:mi:ss)	00:01:12	00:00:29	00:00:17

**Table 5 sensors-20-01848-t005:** Data block size and performance impacts.

	2 KB	4 KB	8 KB	16 KB	32 KB
**Index access costs**	388	390	396	398	400
**Data access using ROWID costs**	880	932	983	1045	1077
**Processing time (hh:mi:ss)**	00:00:58	00:00:59	00:01:02	00:01:03	00:01:03

**Table 6 sensors-20-01848-t006:** Restricted memory structure size and performance impacts.

	2 KB	4 KB	8 KB	16 KB	32 KB
**Index access costs**	397	400	400	402	406
**Data access using ROWID costs**	1278	1386	1615	1630	1640
**Processing time (hh:mi:ss)**	00:01:06	00:01:02	00:01:08	00:01:04	00:01:04

**Table 7 sensors-20-01848-t007:** Operation costs.

	2 KB	4 KB	8 KB	16 KB	32 KB
**Data access using Rowid costs**	1278	1386	1615	1630	1640
**Operation 1**	100	114	134	140	140
**Operation 2**	23	34	40	40	41
**Operation 3**	415	429	485	358	189
**Operation 4**	740	809	956	1092	1270

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
