# Peer review of "Data Block and Tuple Identification Using Master Index"

_sensors, 2020, doi:10.3390/s20071848_

Round 1
Reviewer 1 Report
The article addresses the important issue of optimizing data search in relational databases. These databases often require full searches that are resource-intensive. Indices have been used for this purpose for several decades. The authors propose their solution in this field. At the same time, there is no comparison with current research in this field. The tests are carried out on a DBMS Oracle 11g for which the producer's support ends this year. And modern databases are increasingly stored in the cloud and solutions should be sought to optimize cloud operations, not disk operations.
The scientific value should be more emphasized. Comparison of cloud solutions could also be mentioned. Future orientation behind the Oracle 11g is focused on which environments?
Reviewer 2 Report
This paper presents a comprehensive analysis of some important aspects of data block management when working with sensor data. Authors proposed a couple of alternative improvements in data block location and identification, experimentally analyzed and compared them, including default settings. They showed clear added value in terms of the overall performance of some of the proposed approaches. Moreover, analysis of block size settings in terms of relevant database operations has been performed and evaluated as well.
The authors present an excellent introduction to the researched topic. The description and experimental evaluation of the proposed methods are also very well. A weaker point is the reference to other similar works. Authors address some relevant results in the are within the sections 2 and 3, but no clear comparison of their proposed approaches to other state-of-the-art approaches is provided.
Some specific comments and suggestions:
I recommend using unique names for particular own approaches presented and evaluated. It is quite confusing to see the term model 1 used for two different methods.
In Figure 7, one arrow has the wrong direction - alternative "no" should be outgoing from the decision block "reached HWM?".
Line 419 - it should be "reflected in the physical data".
Line 427 - "it is" should be omitted.
Line 490 - Fig. 12 should be referenced.
Page 18 at the bottom - Results for models M3a and M3b are not commented.
Lines 819 and 820 - presented values are not corresponding to the ones which are provided in Tab. 3.
Line 850 - there should be "data to be processed originate".
Reviewer 3 Report
I am not familiar with Oracle, so I assume what they describe is possible.
In lines 328 to 339 they describe the system they used. In addition, I would like to know the operating system used, the structure of the table and which fields are the primary key.
In lines 340ff it would be interesting to be told how each Model looks like for the example table and query.
In the Data Block discussion (lines 682ff) I would like to know how many rows of the example table fit in a block depending on block size.
Minor issues:
Fig. 5: Should it not say "no" between master index and TAF method?
line 427,428: not a sentence.
line 467: last too long instead of much
Reviewer 4 Report
This paper presents a kind of index oriented access method for the table in the database systems. It provides abundant testing results to prove the proposed "Master index scheme".
The main problem of this paper is that it is nothing new. The same method is already used in many database systems. For example, Oracle already provides the same idea called Index Organized Table. Similarly, MySQL's table is all index-organized table and it is exactly doing what proposed in the paper.
The second problem is that empty blocks in the middle of tablespace can be removed by doing "compaction" or "vacuum". In the real environment, the fragmentation ratio is monitored and compact tables regularly.
The third issue is the evaluation environment. HDD is used for the main storage, so the number of IO would be matter. However, these days flash-based SSD is widely used and HDD is only used for archive data which usually for sequential access. Thus it used HDD, the performance improvement can be amplified.
Round 2
Reviewer 1 Report
The article addresses the important issue of optimizing data search in relational databases. These databases often require full searches that are resource-intensive. Indexes have been used for this purpose for several decades. The authors propose their solution in this field. They also show a comparison with current research in this field. The tests are carried out on the latest DBMS Oracle 19, which also shows current solutions and their quality.